# Does daily activity overlap of seven mesocarnivores vary based on human development?

**Leah E. McTigue**[1]*, **Ellery V. Lassiter**[1], **Mike Shaw**[1], **Emily Johansson**[1], **Ken Wilson**[1], **Brett A. DeGregorio**[2]

1 Department of Biological Sciences, University of Arkansas, Fayetteville, AR, United States of America, 2 U. S. Geological Survey, Arkansas Cooperative Fish and Wildlife Research Unit, Fayetteville, AR, United States of America

* lmctigue@uark.edu

**Data Availability Statement:** All Data is publicly available through Snapshot USA. Publications of data used are as follows: Cove MV, Kays R,

## Abstract

Many species of wildlife alter their daily activity patterns in response to co-occurring species as well as the surrounding environment. Often smaller or subordinate species alter their activity patterns to avoid being active at the same time as larger, dominant species to avoid agonistic interactions. Human development can complicate interspecies interactions, as not all wildlife respond to human activity in the same manner. While some species may change the timing of their activity to avoid being active when humans are, others may be unaffected or may benefit from being active at the same time as humans to reduce predation risk or competition. To further explore these patterns, we used data from a coordinated national camera-trapping program (Snapshot USA) to explore how the activity patterns and temporal activity overlap of a suite of seven widely co-occurring mammalian mesocarnivores varied along a gradient of human development. Our focal species ranged in size from the large and often dominant coyote (*Canis latrans*) to the much smaller and subordinate Virginia opossum (*Didelphis virginiana*). Some species changed their activity based on surrounding human development. Coyotes were most active at night in areas of high and medium human development. Red fox (*Vulpes vulpes*) were more active at dusk in areas of high development relative to areas of low or medium development. However, because most species were primarily nocturnal regardless of human development, temporal activity overlap was high between all species. Only opossum and raccoon (*Procyon lotor*) showed changes in activity overlap with high overlap in areas of low development compared to areas of moderate development. Although we found that coyotes and red fox altered their activity patterns in response to human development, our results showed that competitive and predatory pressures between these seven widespread generalist species were insufficient to cause them to substantially alter their activity patterns.

Bontrager H, Bresnan C, Lasky M, Frerichs T, Klann R, Lee Jr TE, Crockett SC, Crupi AP, Weiss KC. SNAPSHOT USA 2019: a coordinated national camera trap survey of the United States. 2021 Apr 01; 102(6):e03353 Kays R, Cove MV, Diaz J, Todd K, Bresnan C, Snider M, Lee Jr TE, Jasper JG, Douglas B, Crupi AP, Weiss KC. SNAPSHOT USA 2020: A second coordinated national camera trap survey of the United States during the COVID-19 pandemic. Ecology. 2022 Jun 05; 103(10):e3775.

**Funding:** The authors received no specific funding for this work.

**Competing interests:** The authors have declared that no competing interests exist.

## Introduction

Interspecific interactions are key drivers of the structure of wildlife communities with both competition and predation influencing the distribution, behavior, activity patterns, and resource use of co-occurring species [1–3]. Because many species of mammal are capable of being active during the day or night [4, 5], they may shift their behavior to maximize access to resources and avoid competitors or predators [6]. Both competition or the threat of predation can cause shifts in behavior such that subordinate or smaller species may choose to be active when dominant species are least active [7, 8]. Conversely, dominant species should seek to align their activity patterns with those of their prey to optimize energy gain or be active when conditions are optimal without the fear of competitors [9]. From a competitive standpoint, smaller-bodied or subordinate species may be active during times when larger or more dominant species are least active to avoid agonistic interactions [6]. Explorations of activity overlap have provided important insights into how wildlife species behaviorally respond to their environment, competitors, and predators [6, 7, 10, 11]. However, the activity patterns of wildlife are driven not only by co-occurring species but may be strongly influenced by anthropogenic disturbances and human activity [4, 7, 12].

Many species of wildlife have adapted to co-occur with humans in developed areas (i.e. urban adapted species) [13] and in these areas their temporal activity patterns may differ from counterparts in more natural settings [12, 14–16] When co-occurring with humans, many species respond by becoming active when humans are least active, typically at night [4, 17–21]. This may be particularly true for large mammalian species that are often disturbed by humans [17, 21–23] and thus sensitive to human activity and development. If co-occurring species respond similarly to humans by shifting to a more nocturnal existence, then the activity overlap between species may necessarily be higher in developed areas than in undeveloped areas [7, 20]. However, not all urban-adapted wildlife species respond to human presence in the same manner even within taxa groups such as mesocarnivores (medium-sized mammalian predators [7, 24]). Thus, species that are more tolerant of human activity may not switch to a nocturnal existence to avoid humans; thereby reducing interactions with competitors or predators that do so [7, 25]. Furthermore, changes in activity patterns may vary based on the intensity of human development such that changes in activity patterns may vary along a gradient of development [7, 14, 15].

In human-dominated landscapes where larger-bodied mesocarnivore activity may be constrained temporally by an avoidance of humans, subordinate species may alter their activity patterns to avoid overlap, a phenomenon referred to as "development-mediated temporal avoidance" [20, 26, 27]. In situations where the overlap in activity of dominant and subordinate competitors are reduced or altered due to the presence of humans or development, subordinate species may be shielded by development [20, 28]. In some cases, interference competition by larger bodied or dominator competitor species may be reduced in human developed areas [29, 30].

Mesocarnivore communities offer an excellent opportunity to study these interactions, because they have adapted to co-occur in human dominated landscapes and are generally well-studied [31–33]. In North America, the coyote (*Canis latrans*) is the largest mesocarnivore occurring regularly in both natural and human developed ecosystems [34] and is considered dominant over many co-occurring, smaller-bodied mesocarnivore species that also regularly occur across natural and developed areas, such as northern raccoons (hereafter raccoon: *Procyon lotor*), striped skunk (hereafter skunk, *Mephitis mephitis*), red fox (*Vulpes vulpes*), gray fox (*Urocyon cinereoargenteus*), bobcat (*Lynx rufus*), and Virginia opossum (hereafter, opossum: *Didelphis virginiana* [35, 36]). Each of these species overlap widely in

geographic range across much of North America and occupy habitats along a gradient of human development [37, 38]. Many of these mesocarnivores hunt and compete for the same prey [39]. Additionally, raccoons, skunks, gray fox, and opossums not only compete with larger mesocarnivores like the coyote and bobcat but are also vulnerable to predation or killing [35, 40, 41].

Here, our overall objectives were to evaluate whether human development causes changes in the activity patterns and activity overlap of a suite of co-occurring mesocarnivores. Using camera trap data from a nationwide collaborative effort (Snapshot USA 11, 38]) our specific objectives were to 1) characterize the activity patterns of mesocarnivore species at differing levels of human development, 2) evaluate the degree of temporal activity overlap between species pairs, and 3) determine if the degree of surrounding human development correlates with changes in activity overlap between species pairs. We predicted that all of the focal mesocarnivore species would primarily be nocturnal and thus temporal activity overlap between species pairs would generally be high. However, we predicted that in areas of human development the largest mesocarnivore species would be most sensitive to human activity and thus become almost exclusively nocturnal which would result in lower levels of overlap with small-bodied mesocarnivores that would not be as human-intolerant.

## Methods

### Data selection

We used camera trap data from Snapshot USA [42, 43]. Snapshot USA is a nationwide coordinated camera trapping project in which contributors deploy motion-triggered infrared cameras annually in a standardized manner between September 1st and October 31st. At each study site, each contributor deploys at least 8 cameras in a single type of habitat (forest, grassland, desert, anthropogenic, etc, S1 Table) and aims to collect a minimum of 400 trap nights. Cameras are placed at least 200m apart, but no more than 5km apart. If sites were sampled both years, they were treated as individual sites for analysis.

To characterize how human development surrounding each Snapshot USA site, we created 500m buffers around each camera using ArcGIS (ArcGIS Pro; Esri Inc, Redlands, CA). We chose 500m buffers because this spatial scale has been shown to be relevant to the occurrence and behavior of common North American mesocarnivores [7, 33, 44]. Within the 500m buffer we calculated the amount of landcover categorized as human development, (the combination of low, medium, and high-intensity development as measured by the 2019 National Land Cover Database [45]). We then averaged this development value for all cameras at a given site. We then binned each site into one of three development categories: undeveloped sites (0–1% development), moderately developed (1–10% development), or heavily developed (10–100% development).

### Focal species selection

To select focal study species, we explored the complete Snapshot USA 2019 and 2020 datasets [42, 43]. We isolated all species of carnivores and mesocarnivores detected (N = 29). We then filtered this list based on several criteria that we deemed essential for inclusion in this investigation. First, we only included species that occurred within each of the three development categories (highly developed, moderately developed, or low human development) because this allowed us to explore changes in activity patterns across the human development gradient. Next, we needed to include criteria pertaining to sample sizes to ensure we had both statistical power to make comparisons between development categories as well as enough detections of a species to quantify its activity pattern (see below in *Activity Patterns and Overlap*). Therefore,

we only used species that occurred at a minimum of 5 study sites within each development category as this allowed us to create overlap means for use in analyses. Finally, at each site that a species was detected, there needed to be a minimum of 2 detections at that site in order for us to calculate activity kernels. Based on these criteria, we narrowed our focus to seven species of mesocarnivore including Northern raccoon, striped skunk, red fox, gray fox, bobcat, coyote, and Virginia opossum. Collectively, these species occurred at 210 Snapshot USA sites sampled in either 2019 or 2020 (Fig 1). Of these 210 sites, 44 were categorized as high development, 47 were medium development, and 119 were low development.

### Activity patterns and overlap

For each detection of a focal species, we extracted the date and time of the detection. We then determined if the detection occurred during the day, night, around dusk, or around dawn by calculating location and date-specific sunset and sunrise times using the R package sunCalc [46]. Detections occurring within an hour of sunset were categorized as dusk, and detections occurring within an hour of sunrise were categorized as dawn. Nocturnal events (classified as night) occurred after dusk and before dawn. Diurnal events (classified as day) occurred after dawn and before dusk. At each site, we calculated the proportion of events occurring during each of the four daily activity periods (dawn, dusk, day, or night). We compared the proportion of events occurring during each of the four time periods across three development groups (high development, medium development, and low development) using a Kruskal-Wallis test. We then used Dunn tests with Bonferroni corrections to make pair-wise comparisons between the development categories. We used non-parametric tests due to non-normality in the data.

To calculate the activity overlap for each of the possible pair-wise comparisons of our seven focal species, we used the R package overlap [47]. We first converted the time of each detection to radians to account for the circular nature of time. We then calculated a site and species-specific non-parametric kernel around the detection times to quantify when each species was most active during the diel period [48]. For each site, we calculated the area under the kernel curve that overlapped between each of the species pairs, which was quantified using a 0–1 index with 0 indicating no overlap and 1 indicating complete temporal overlap [49]. We used $\Delta_1$ for small sample sizes because most sites had fewer than 75 detections per species [48, 49]. We excluded sites where detections were too few to generate kernel density estimates. This approach generated an overlap coefficient for each species pair for each study site. We then compared the mean overlap coefficients between each species pair across the three development groups using Kruskal-Wallis and Dunn tests.

## Results

We included data from a total of 210 Snapshot USA sites spanning the 2019 and 2020 seasons. Sites that were sampled in both years were considered separate sites for our analyses because camera locations varied between years. Collectively, these 210 study sites accumulated 106,467 total trap nights and 38,076 detections of the seven focal species. Each species had at least 800 overall detections and was detected at a minimum of 47 sites (Table 1).

We found that all seven species that were predominantly nocturnal were occasionally active during day, dawn, and/or dusk (Fig 2). Opossum were the most strongly nocturnal species ( ± *SE*: 96.64 ± 0.77) followed by striped skunk (92.12 ± 2.87), raccoon (88.32 ± 1.828), gray fox (85.92 ± 5.92), red fox (80.58 ± 4.92), coyote (69.5 ± 2.78), and bobcat (60.05 ± 5.66).

We found that both coyote and red fox activity changed in response to human development (Fig 2). The proportion of nocturnal coyote detections increased at sites with medium human development compared to areas of low development ($X^2$ = 7.66, *df* = 2, *P* = 0.02), though there

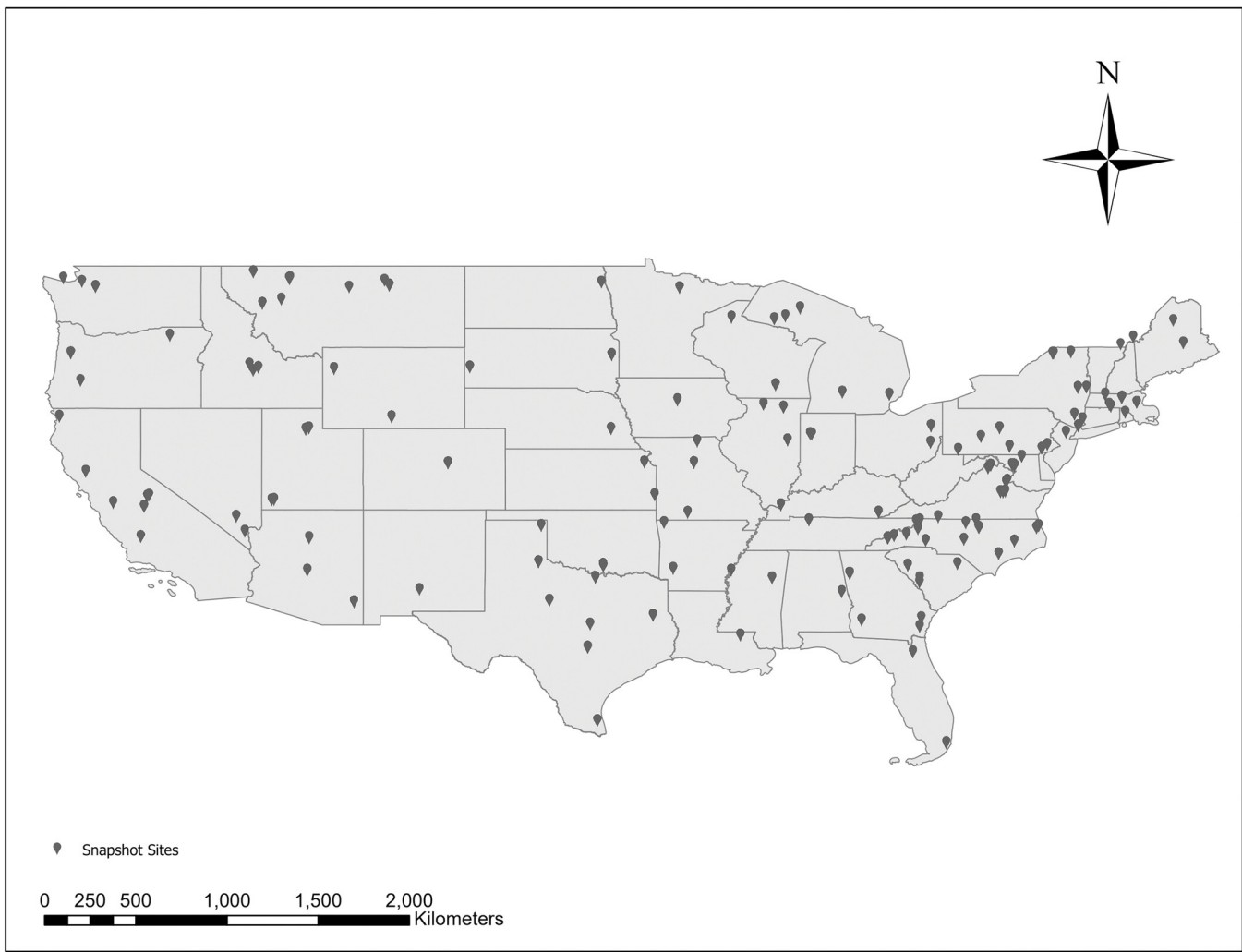

**Fig 1. Snapshot USA sites: 2019 and 2020.** Camera trap sites from Snapshot USA 2019 and 2020 (USGS National Boundary Dataset (NBD)). The 210 sites used were determined though selection of 7 species of mesocarnivores where 1) each species occurred in at least 5 sites of each development category (High (44), Medium (47), Low (119)); and 2) had at least 2 detections at each site.

**Table 1. Detection summary for seven mesocarnivores from Snapshot USA.**

| Species | No. Sites with Detections | No. Overall Detections |
|---|---|---|
| **Bobcat** | 85 | 877 |
| **Coyote** | 184 | 5708 |
| **Gray Fox** | 47 | 854 |
| **Red Fox** | 73 | 3264 |
| **Raccoon** | 163 | 20801 |
| **Skunk** | 63 | 804 |
| **Opossum** | 128 | 5768 |

Total number of detections of seven focal mesocarnivore species from a nationwide camera trapping study Snapshot USA 2019 and 2020 that were used to explore changes in activity patterns and activity overlap in relation to human development.

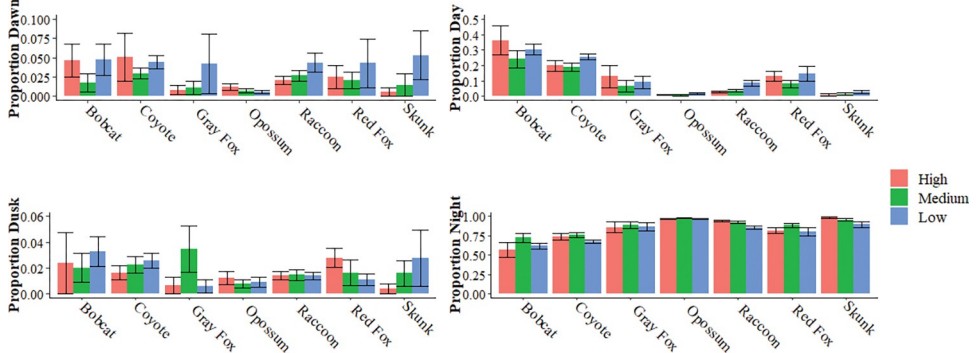

**Fig 2. Daily Activity Patterns for seven mesocarnivores along an urban to rural gradient.** The proportion of activity of seven common mesocarnivores during different diel periods based on surrounding levels of development as measured by the 2019 National Land Cover Database (NLCD). Derived from data collected across 210 study sites across the United States as part of Snapshot USA 2019 and 2020 [42, 43]. Values are means ± SE.

was no difference between medium and high development. The proportion of red fox detections occurring around dusk was greatest in areas with high development ($X^2 = 11.54$, $df = 2$, $P < 0.01$).

In general, temporal overlap between each species pair was high with most species' pairs having temporal overlap values of over 0.70 (Table 2). Variation in overlap based on development was only observed between opossum and raccoon (Fig 3). In areas of high development, overlap was 0.81 ± 0.09 (mean± 1 SD). Overlap was lowest in areas with medium development (0.76 ± 0.01, but highest in low development (0.82 ± 0.08), indicating the two species shift the timing of their activity as urbanization increases from low to medium, but were unaffected between medium and high development ($X^2 = 4.72$, $df = 2$, $P = 0.10$; Fig 3). We found no significant differences in overlap between the other species pairs based on development (Table 2). Red fox and gray fox only overlapped in areas of high and low development.

## Discussion

There is extensive evidence that many species of wildlife alter their activity patterns when in the presence of humans or human development [4, 7, 50]. However, there has been less attention paid to how these changes alter interspecies interactions [25]. We predicted that human development could either increase overlap in the timing of mesocarnivore activity by constraining activity to periods when animals avoided interactions with humans or it could reduce overlap between some species as those that are more tolerant of humans would be able to be active during daytime whereas others were more sensitive and would only be active at night [25]. However, we found relatively little support for these predictions. The seven species we studied were primarily nocturnal regardless of surrounding human development and thus, overlap between the species pairs was consistently high (Table 2). Overall, we saw little to no change in overlap between species on an urban to rural gradient. The only significant change in overlap occurred between raccoon and opossum, which had a higher overlap coefficient in sites of low development than those with medium development. We had predicted that larger mesocarnivores that are at risk of disturbance from humans would be most nocturnal in developed areas, which was the case for coyotes. Similarly, red fox were more active at dusk when in proximity to human development.

The mesocarnivore species studied here were primarily nocturnal in their activity patterns as has been reported in the literature [5, 7, 51–54]. However, each species was plastic in their

**Table 2. Temporal overlap coefficients for seven mesocarnivores across the United States.**

| Species Pair | Species Pair Code | No. Overlapping Sites | No. Development Categories | High | | Medium | | Low | |
|---|---|---|---|---|---|---|---|---|---|
| | | | | $\Delta_1$ | SD | $\Delta_1$ | SD | $\Delta_1$ | SD |
| Coyote/Bobcat | CB | 107 | 3 | 0.81 | 0.10 | 0.77 | 0.09 | 0.74 | 0.11 |
| Coyote/Gray Fox | CG | 56 | 3 | 0.72 | 0.16 | 0.71 | 0.14 | 0.74 | 0.08 |
| Coyote/Raccoon | CN | 158 | 3 | 0.76 | 0.08 | 0.74 | 0.12 | 0.76 | 0.11 |
| Coyote/Opossum | CO | 124 | 3 | 0.73 | 0.14 | 0.73 | 0.11 | 0.76 | 0.10 |
| Coyote/Red Fox | CR | 75 | 3 | 0.71 | 0.16 | 0.71 | 0.08 | 0.78 | 0.10 |
| Coyote/Skunk | CS | 84 | 3 | 0.64 | 0.19 | 0.74 | 0.08 | 0.75 | 0.06 |
| Gray Fox/Bobcat | GB | 37 | 3 | 0.63 | 0.08 | 0.71 | 0.13 | 0.76 | 0.09 |
| Gray Fox/Red Fox | GR | 24 | 2 | 0.74 | 0.14 | - | - | 0.82 | 0.09 |
| Gray Fox/Skunk | GS | 30 | 3 | 0.50 | 0.14 | 0.75 | 0.19 | 0.77 | 0.11 |
| Raccoon/Bobcat | NB | 97 | 3 | 0.69 | 0.14 | 0.71 | 0.10 | 0.73 | 0.13 |
| Raccoon/Gray Fox | NG | 52 | 3 | 0.71 | 0.10 | 0.76 | 0.16 | 0.79 | 0.10 |
| Raccoon/Red Fox | NR | 82 | 3 | 0.70 | 0.13 | 0.72 | 0.10 | 0.77 | 0.11 |
| Raccoon/Skunk | NS | 71 | 3 | 0.75 | 0.10 | 0.72 | 0.11 | 0.78 | 0.11 |
| Opossum/Bobcat | OB | 75 | 3 | 0.65 | 0.14 | 0.69 | 0.11 | 0.72 | 0.15 |
| Opossum/Gray Fox | OG | 42 | 3 | 0.77 | 0.09 | 0.68 | 0.15 | 0.77 | 0.07 |
| Opossum/Raccoon | ON | 134 | 3 | 0.81 | 0.09 | 0.76 | 0.10 | 0.82 | 0.08 |
| Opossum/Red Fox | OR | 63 | 3 | 0.74 | 0.12 | 0.73 | 0.14 | 0.78 | 0.09 |
| Opossum/Skunk | OS | 55 | 3 | 0.81 | 0.10 | 0.79 | 0.08 | 0.74 | 0.11 |
| Red Fox/Bobcat | RB | 39 | 3 | 0.62 | 0.17 | 0.60 | 0.13 | 0.74 | 0.11 |
| Red Fox/ Skunk | RS | 40 | 3 | 0.80 | 0.04 | 0.73 | 0.14 | 0.73 | 0.09 |
| Skunk/Bobcat | SB | 54 | 3 | 0.61 | 0.27 | 0.69 | 0.13 | 0.72 | 0.11 |

Temporal overlap coefficients ($\Delta_1$) for seven commonly occurring mesocarnivore species: bobcat (*Lynx rufus*), raccoon (*Procyon lotor*), Virginia opossum (*Didelphis virginiana*), striped skunk (*Mephitis mephitis*), gray fox (*Urocyon cinereoargenteus*), red fox (*Vulpes vulpes*), and coyote (*Canis latrans*). Development categories refer to how many categories of development (low, medium, high) each species pair was detected in and were included in analyses. Gray and red fox only overlapped in two of the three development categories. Overlapping sites indicate the total number of Snapshot USA sites where both species in the pair were detected. Overlap coefficients and standard deviation range from 0–1 with 0 indicating no temporal overlap between the activity of the two species and 1 indicating complete temporal overlap.

activity and did show the ability to be active during each of the diel activity periods (Fig 2). We expected that some of the species studied here would substantially change the timing of their activity based on human development as has been reported elsewhere [4, 7, 54, 55]. For example, both raccoons and opossums shifted activity to crepuscular hours when in proximity to humans [54]. Coyotes have been found to avoid being active during the day and to shift to nocturnal activity in the presence of human activity [7, 16, 23, 25, 32]. Other species, like striped skunk and bobcat have shown to be less flexible in their activity patterns in regard to development or to respond to environmental factors such as temperature rather than humans [7, 24, 25]. We found only minor changes in the activity patterns of our focal species across the development categories. For instance, we found that coyotes became more nocturnal in areas of human development which aligns with other reports and that red fox became more crepuscular in areas of human development (Fig 2).

We may have detected less change in activity patterns than anticipated because our activity categories (day, night, dawn, dusk) were too broad, and the species we focused on respond to human activity by shifting the times they are active within those categories rather than across them. For instance some mammal species constrain their activity to fewer hours of the night in urban areas rather than shifting when they are active across broad categories [54] (e.g., crepuscular to nocturnal). Alternatively, our categorization of human development may not perfectly

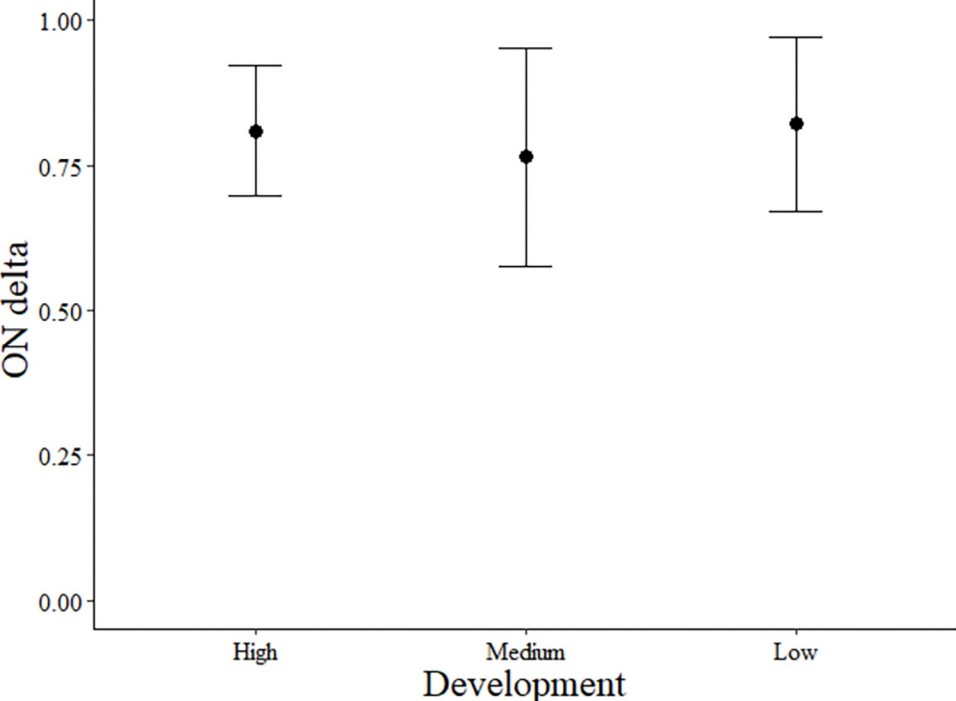

**Fig 3. Temporal overlap patterns between Virginia opossum (*Didelphis virginiana*) and northern raccoon (*Procyon lotor*) along an urban to rural gradient.** Daily activity overlap between Virginia opossum and northern raccoon (ON). We compare mean overlap coefficient and standard deviation values at three levels of development as measured by the 2019 National Land Cover Database (NLCD).

capture the aspects of human impact that wildlife respond to. We relied upon a measure of impervious surface [45] within the vicinity of each site to categorize human development. The use of impervious surface is a common measure of human development in studies of wildlife response to humans [7]. However, wildlife may be responding to other aspects of human presence such as anthropogenic noise, human activity, traffic, domestic dogs, or light that may not correlate with impervious surface [23, 56].

We found that overlap between most species' pairs was high (overlap coefficient > 0.70), likely because all seven of the species we studied were mostly nocturnal (Table 2). Furthermore, we found that contrary to our predictions, overlap remained consistently high regardless of the amount of human development surrounding sites. This consistent temporal overlap between species pairs indicate either that 1) competition / predation pressure between these species pairs is not strong enough to induce substantial shifts in activity to avoid one another or 2) that these species use different mechanisms to reduce competition and predation risk. We expected that pressure from larger mesocarnivores such as coyote and bobcat would be enough for smaller mesocarnivores like opossum and skunk to change their activity patterns because these species have been documented killing and depredating these species (up to 76% mortality of opossum due to coyote [35, 57]. Therefore, we expected that these mesocarnivores should benefit by avoiding when coyotes or bobcat were most active [21, 24]. Furthermore, interspecific aggression and killing has been documented between the larger mesocarnivores, including coyote, bobcat, gray fox, and red fox [58, 59]. In addition to antagonistic encounters, these four species undoubtedly compete for similar prey resources [60]. Despite evidence of competition, predation, and antagonistic behavior, we found no evidence that these species

alter their activity patterns to reduce temporal overlap. Rather, it is more likely that they reduce contact through some combination of spatial or spatio-temporal partitioning as has been reported in other systems [20].

Due to the scale and scope of Snapshot USA, there was wide variation in the type and intensity of human development at study sites, and this may have contributed to our relative lack of evidence for consistent shifts in behavior. The way in which wildlife respond to human development are often context dependent [61, 62] and wildlife responses to development may vary regionally or in response to the characteristics of that specific development as well as the predominant landcover. While the majority of Snapshot USA sites were in forested areas (S1 Table) other sites were located in desert, grassland, or wetland areas. Our broad approach likely missed much of the nuance that would have contributed to a better understanding of how wildlife respond to their environment, and we lacked the statistical power to evaluate the effects of habitat on activity. Due to the modest shifts in the activity of our focal species, we failed to detect changes in overlap between the species based on the surrounding human development. Future research could benefit from including focal species with even more plasticity in the timing of their activity (e.g., nine-banded armadillos (*Dasypus novemcinctus*), [56]) as opposed to species that are consistently nocturnal or by focusing on more specialized predator–prey models rather than the generalist species we focused on.

## Supporting information

**S1 Table. Habitat at each of the Snapshot USA camera sites used in analyses.**
(DOCX)

**S1 File.**
(ZIP)

## Acknowledgments

Thank you to Roland Kays and the Snapshot USA team for access to data and assistance in refining our research question. Thanks to the University of Arkansas Dept of Biological Sciences for student support. Any use of trade, firm, or product names is for descriptive purposes only and does not imply endorsement by the U.S. Government. Data are available the Smithsonian's eMammal data repository https://emammal.si.edu/analysis/data-download.

## Author Contributions

**Conceptualization:** Leah E. McTigue, Brett A. DeGregorio.

**Data curation:** Leah E. McTigue, Ellery V. Lassiter, Mike Shaw, Emily Johansson, Brett A. DeGregorio.

**Formal analysis:** Leah E. McTigue.

**Methodology:** Leah E. McTigue, Brett A. DeGregorio.

**Project administration:** Leah E. McTigue, Brett A. DeGregorio.

**Supervision:** Brett A. DeGregorio.

**Validation:** Leah E. McTigue, Brett A. DeGregorio.

**Visualization:** Leah E. McTigue.

**Writing – original draft:** Leah E. McTigue, Brett A. DeGregorio.

**Writing – review & editing:** Leah E. McTigue, Ellery V. Lassiter, Mike Shaw, Emily Johansson, Ken Wilson, Brett A. DeGregorio.

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
