## [Decision Letter · Decision Letter 0]

24 Feb 2023

PONE-D-22-35736

Does daily activity overlap of three mesocarnivores vary based on human development?

PLOS ONE

Dear Dr. unit,

Thank you for submitting your manuscript to PLOS ONE. After careful consideration, we feel that it has merit but does not fully meet PLOS ONE’s publication criteria as it currently stands. Therefore, we invite you to submit a revised version of the manuscript that addresses the points raised during the review process.

We look forward to receiving your revised manuscript.

Kind regards,

Carlos Rouco, PhD

Academic Editor

PLOS ONE

Journal Requirements:

“Thanks to the University of Arkansas Dept of Biological Sciences for student support and to Arkansas Game and Fish Commission for funding this project under cooperative agreement No. 1434-04HQRU1567.”

“The authors received no specific funding for this work.”

Additional Editor Comments:

Dear Authors,

I agree with the two reviewers that the MS can be improved. To ensure the Editor and Reviewers will be able to recommend that your revised manuscript is accepted, please pay careful attention to each of the comments that have been pasted underneath this email, and in particular those ised by Reviwer 1. This way we can avoid future rounds of clarifications and revisions, moving swiftly to a decision.

Reviewers' comments:

Reviewer's Responses to Questions

**Comments to the Author**

1. Is the manuscript technically sound, and do the data support the conclusions?

Reviewer #1: Partly

Reviewer #2: Yes

2. Has the statistical analysis been performed appropriately and rigorously? 

Reviewer #1: Yes

Reviewer #2: Yes

3. Have the authors made all data underlying the findings in their manuscript fully available?

Reviewer #1: Yes

Reviewer #2: No

4. Is the manuscript presented in an intelligible fashion and written in standard English?

Reviewer #1: Yes

Reviewer #2: Yes

5. Review Comments to the Author

Reviewer #1: PLoS ONE Review:

I thank the editor and authors for the opportunity to review this manuscript. I hope the authors find my review both helpful and constructive. I’ve organized this review by section, where I first give my general comments followed by more-specific line comments.

In this study, the authors investigate whether the temporal overlap of three mesocarnivores (coyote, Virginal opossum, and northern raccoon) varies in relation to human development by utilizing the large Snapshot USA datasets from 2019 and 2020. In general, they find that although species-specific responses to development occur, the majority of activity for all three species remains relatively nocturnal and the resultant overlap between species pairs does not significantly change across development levels. The authors conclude that competitive or predatory pressures between the three species are insufficient in eliciting a behavioral response across a gradient of human development.

In general, I very much enjoyed this manuscript, and the merit for such a study is obvious. The methods and questions were succinct and well-developed, and the paper had a nice flow throughout. However, especially with a dataset as large as Snapshot USA, I feel like the authors could go further, and I urge them to do so in order to publish this work in a higher-impact journal such as this. Specifically, I’m curious why the authors selected only three mesocarnivores from the Snapshot USA dataset. I think more should be included (e.g., bobcat, red fox, grey fox, striped skunk), especially since recent work has shown connections between the species mentioned above and coyotes, in particular (citations provided below). Furthermore, the Snapshot USA dataset includes a decent amount of data on larger apex predators (e.g., wolf, puma, black bear), and I would be extremely curious to include these species in this study as well. Taking a look at how different species assemblages may affect the overlap between species pairs, and whether or not coyote elicit the same behavioral response as something like puma or black bear across human development, while utilizing more of the data that Snapshot has to offer (only 82 of the nearly 4,000 sites across the total dataset were included), would make this publication more relevant to the general readership of PLoS ONE.

ABSTRACT

General Comments

The last sentence speaks to there being insufficient evidence that competitive or predatory pressures produced a behavioral response in the three species’ activity patterns. However, it says nothing about the relation to human development. As the change in activity in relation to human development was a central theme throughout the manuscript, it seems remiss to exclude any reference to it in the concluding marks of the abstract. As the information is currently presented, the relation to human development seems like an afterthought.

INTRODUCTION

General Comments

There is quite a bit of information in this introduction about predator-prey interactions. This seems completely unnecessary, as this is never studied herein and only alluded to once throughout. Furthermore, much of these references allude to the presence of apex predators on prey species, another point that is not hit on in this particular study. I either recommend: a) removing references to predator-prey interactions when unnecessary and replacing it with background information on co-occurrence between sympatric predators, or b) revamping the study like mentioned above to include apex predators and rewording the sections to discuss the impact they have on more subordinate carnivores.

Minor Comments

Lines 64-68: These statements need a references. Recent work (see Green et al., 2022 in Global Ecology and Conservation; Hubbard et al., 2022 in Biodiversity and Conservation; and Allen et al., 2022 in Global Ecology and Conservation) may better place them in the context of current work.

METHODS

General Comments

As discussed above, I feel like the authors should go deeper than they do, especially when utilizing a dataset as rich as the Snapshot USA one (incorporating data from less than 5% of sites seems like a disservice). There are many other mesocarnivores in the U.S. that occupy urban environments and have relationships with coyote and other mesocarnivores (e.g., see Allen et al., 2022 in Diversity in Distributions, which uses the entire Snapshot dataset). For example, it would be interesting to investigate the link between coyote and red fox, especially in relation to urban development, or the relationship between species such as skunk, raccoon, and red fox, as these species seemingly share very similar ecological niches. Furthermore, by looking at different assemblages of mesopredators across the U.S., the authors can leverage the full power of the Snapshot dataset.

Lastly, descriptive statistics of the sites included in analysis, organized by strata, would be extremely beneficial. How many sites were there in each category? What was the spatial distribution of each category? What climates were included? What was the average precipitation and temperature across sites? Etc.

Minor Comments

Lines 149-154: So Overlap coefficients were calculated on a site-by-site basis? In that case, was the error in site-specific estimates incorporated into the second stage of the analysis? It says that the site-specific confidence intervals were calculated, but I don’t see any reference as to how they were used or propagated though the second stage of modeling. I’m guessing the mean or median estimate from each site was used during the Kruskal-Wallis and Dunn tests. This information should be explicitly specified. Furthermore, if the inherent bias included in using an estimate from one model as an explanatory variable in another model (i.e., doing statistics on statistics) was not explicitly accounted for, than this needs to be mentioned as a caveat in the discussion or later in the methods.

RESULTS

General Comments

All four figures need some work. Specifically, both the x and y axes and the legend are too small, and there is a lot of unnecessary white space. It’s also pretty difficult to discern if any of the plotted differences are statistically significant. If possible, plotting the points as either different colors based on statical differences, or, more appropriately, using a letter designation above each comparison would help clarify this.

There needs to be a map of the study area provided, including sites that were included in analysis. With the current presentation, the reader has no idea where these data were gathered, other than it was across the U.S. A map would help contextualize the extent of these species’ ranges included in analysis, as well as the spatial configuration of sites.

Minor Comments

157-158: Does this mean 82 sites per year or 82 sites across years?

157-159: What was the total sampling effort (i.e., total camera days across the 82 sites)?

DISCUSSION

General Comments

Typically, the first paragraph of the discussion section summarizes the main findings of the study and presents overarching takeaways that are then expanded on in further detail in the ensuing paragraphs. I particularly like this format, as I believe it is the easiest for readers to follow. Although it is not necessary, and is simply a matter of personal opinion, I urge the authors to consider reworking their first paragraph, at least.

Minor Comments

Lines 239-242: This line really reiterates the need for expanding this current study to incorporate the entirety of the Snapshot USA dataset. This is an extremely intriguing question, and identifying where and why species interactions occur or don’t occur would add great value to this work.

Lines 250-252: Again, this seems to signal rationale for this study to expand on this work and look large-scale with the intent of identifying why these differences occur.

Lines 252-255: Can we really say range wide here? Seems like quite a stretch.

Lines 258-260: This variation explained and summarized, either in a supplemental table or in the results section.

Lines 258- 269: I see no reason why this study cannot be the answer to the future research the authors call for.

Reviewer #2: I found this paper interesting, and I think it will make a good contribution to the Plos One journal. I do however have some comments (mostly minor) which I would like addressed. The majority of comments are attached in the document, but the main points I will highlight here;

There is a lack of information in this paper regarding the specific habitats and areas that camera data was collected. I think this is important to summarize, even if only into broad categories. Was the majority of data collected near towns? Agricultural areas? Forests? The use of the housing unit density index alone does not provide any of this info.

I am not convinced that the housing unit density was the best metric to test for temporal differences between species. I would like to see more argument why this was used over other metrics of human development.

Again, following on from the lack of habitat classifications, how common were agricultural areas in this study? In theory these would have low housing density, however meso-carnivores can have particularly high levels of human persecution in these areas, which could impact temporal activity, but not align with the low development idea. I would like to see some clarification on this.

I think it would be good for the authors to create a simple study map of the sites used in the analysis from Snapshot USA. I am curious to see the distribution of this data.

The figures are blurry and need to be improved. I suggest also moving the axis titles further away from the axis.

6. PLOS authors have the option to publish the peer review history of their article (what does this mean?). If published, this will include your full peer review and any attached files.

Reviewer #1: No

Reviewer #2: No

---

## [Author Response · Author response to Decision Letter 0]

12 Jun 2023

Thank you for your thorough and thoughtful review of our manuscript. Detailed responses to the specific reviewer comments can be found in the Reviewer Response Document. We appreciate the opportunity to resubmit our manuscript to PLOS ONE.

---

## [Decision Letter · Decision Letter 1]

28 Jun 2023

Does daily activity overlap of seven mesocarnivores vary based on human development?

PONE-D-22-35736R1

Dear Dr. unit,

We’re pleased to inform you that your manuscript has been judged scientifically suitable for publication and will be formally accepted for publication once it meets all outstanding technical requirements.

Kind regards,

Carlos Rouco, PhD

Academic Editor

PLOS ONE

Additional Editor Comments (optional):

Reviewers' comments:

Reviewer's Responses to Questions

**Comments to the Author**

1. If the authors have adequately addressed your comments raised in a previous round of review and you feel that this manuscript is now acceptable for publication, you may indicate that here to bypass the “Comments to the Author” section, enter your conflict of interest statement in the “Confidential to Editor” section, and submit your "Accept" recommendation.

Reviewer #1: All comments have been addressed

2. Is the manuscript technically sound, and do the data support the conclusions?

Reviewer #1: Yes

3. Has the statistical analysis been performed appropriately and rigorously? 

Reviewer #1: Yes

4. Have the authors made all data underlying the findings in their manuscript fully available?

Reviewer #1: Yes

5. Is the manuscript presented in an intelligible fashion and written in standard English?

Reviewer #1: Yes

6. Review Comments to the Author

Reviewer #1: I'd like to congratulate the authors on a very thorough revision, and I really appreciate the time and effort they spent on addressing my initial concerns. I now strongly feel that the manuscript is ready for publication. Great work!

7. PLOS authors have the option to publish the peer review history of their article (what does this mean?). If published, this will include your full peer review and any attached files.

Reviewer #1: No

---

## [Editor Report · Acceptance letter]

4 Jul 2023

PONE-D-22-35736R1 

Does daily activity overlap of seven mesocarnivores vary based on human development? 

Dear Dr. unit:

I'm pleased to inform you that your manuscript has been deemed suitable for publication in PLOS ONE. Congratulations! Your manuscript is now with our production department. 

Kind regards, 

on behalf of

Dr. Carlos Rouco 

Academic Editor

PLOS ONE